# Anti-Inflammatory Constituents of *Antrodia camphorata* on RAW 264.7 Cells Induced by Polyinosinic-Polycytidylic Acid

**DOI:** 10.3390/molecules27165320

**Published:** 2022-08-20

**Authors:** Ping-Chen Tu, Wen-Ping Jiang, Ming-Kuem Lin, Guan-Jhong Huang, Yi-Jen Li, Yueh-Hsiung Kuo

**Affiliations:** 1Department of Chinese Pharmaceutical Sciences and Chinese Medicine Resources, China Medical University, Taichung 40402, Taiwan; 2Department of Pharmacy, Chia Nan University of Pharmacy and Science, Tainan 71710, Taiwan; 3Department of Food Nutrition and Healthy Biotechnology, Asia University, Taichung 413, Taiwan; 4Department of Nutrition and Health Sciences, Chang Jung Christian University, Tainan 71101, Taiwan; 5Department of Biotechnology, Asia University, Taichung 41354, Taiwan; 6Chinese Medicine Research Center, China Medical University, Taichung 40402, Taiwan

**Keywords:** *Antrodia camphorata*, 4-acetylantroquinonol B, RAW 264.7 cells, poly I:C

## Abstract

*Antrodia camphorata* is an endemic mushroom in Taiwan. This study was designed to screen anti-inflammatory compounds from the methanolic extract of the mycelium of *A. camphorata* on nitric oxide (NO) production in RAW 264.7 cells induced by polyinosinic-polycytidylic acid (poly I:C), a synthetic analog of double-stranded RNA (dsRNA) known to be present in viral infection. A combination of bioactivity-guided isolation with an NMR-based identification led to the isolation of 4-acetylantroquinonol B (**1**), along with seven compounds. The structure of new compounds (**4** and **5**) was elucidated by spectroscopic experiments, including MS, IR, and NMR analysis. The anti-inflammatory activity of all isolated compounds was assessed at non-cytotoxic concentrations. 4-Acetylantroquinonol B (**1**) was the most potent compound against poly I:C-induced NO production in RAW 264.7 cells with an IC_50_ value of 0.57 ± 0.06 μM.

## 1. Introduction

Inflammation is part of the innate, or nonspecific, immune responses against invading pathogens. It is generally initiated when pathogen-associated molecular patterns (PAMPs), such as pathogen-specific carbohydrates, lipoproteins, and nucleic acids, are recognized by a large family of pattern-recognition receptors (PRRs), expressed by macrophages, monocytes, dendritic cells, and neutrophils [1]. Among these immune cells, macrophages play a crucial role in inflammatory reaction through producing inflammatory factors, such as reactive oxygen species (ROS), nitric oxide (NO), cytokines, chemokines, and prostaglandins [2,3].

Notable outbreaks in the last two decades mostly involved respiratory viruses—such as the 2003 severe acute respiratory syndrome (SARS) epidemic, 2009 A(H1N1) pandemic, Middle Eastern respiratory syndrome (MERS) epidemic, and ongoing coronavirus disease 2019 (COVID-19) pandemic—that represent a considerable challenge to worldwide public health [4]. Novel respiratory viruses commonly lead to severe respiratory tract infections. Owing to the involvement of inflammatory signaling pathways in imbalanced host immune response, which cause acute lung injury in respiratory virus infection, mounting evidence points to the therapeutic potential of targeting viral induced-inflammation [5,6,7].

Plants are generally considered a rich source of natural products that can prevent the onset of diseases and reduce health care costs. A variety of natural products, including flavonoids, terpenoids, alkaloids, glycosides, quinones, and phenolic compounds, from plants traditionally used as antiviral agents have been reported to possess remarkable potential against viral infection [8,9,10]. Among them, anti-inflammatory phytochemicals have attracted much attention to providing potential therapeutic interventions in combating viruses and related complications [11]. *Antrodia camphorata* is an edible and precious mushroom endemic to Taiwan, which has been used as traditional medicine. *A. camphorata* has been found to exhibit a broad range of pharmacological effects, which include anti-microbial, anti-oxidative, anti-inflammatory, anti-diabetic, anti-aging, anti-carcinogenic, neuroprotective, hepatoprotective, cardioprotective, and immunomodulatory effects [12,13,14,15]. Phytochemical investigation of *A. camphorata* led to the isolation and characterization of diverse bioactive phytochemicals that include benzenoids, ubiquinones, maleic anhydride and maleimide derivatives, terpenoids, and steroids [12,13,14,15]. Antcins, unique steroid-like compounds isolated from *A. camphorata*, were reported to be novel anti-angiotensin-converting enzyme 2 (ACE2) agents for prophylaxis of COVID-19 [16]. Whether anti-inflammatory phytochemicals from *A. camphorata* show potency against inflammation in viral infection remains to be fully elucidated.

Recently, metabolomics can provide a rapid tool by allowing mixture analysis and simultaneously detecting diverse phytochemicals without the need for time-consuming chromatographic procedures. NMR-based identification is extremely powerful for rapid identification of natural products with relative concentration determination. Furthermore, relating metabolites with bioactivities is feasible by integrating metabolite profiling with bioassay results [17,18]. In this study, we combined an NMR-based identification approach with an anti-inflammatory assay to select natural products with inhibitory effects on NO production in RAW 264.7 cells induced by poly I:C, a synthetic analog of dsRNA known to be produced by viral replication and presented in viral infection.

## 2. Results and Discussion

Viral infections might provoke hyper-inflammation responses in macrophages. dsRNAs are produced by viral replication during viral infections and recognized by PPR [19]. As a synthetic analog of dsRNA, poly I:C was used to mimic the viral infection model both in vivo and in vitro in previous studies [19,20,21].

The inhibitory effects of the mycelium of *A. camphorata* on viral infection-associated inflammation have not been studied previously. Here, poly I:C was applied to stimulate NO production in a RAW 264.7 mouse macrophage cell line as a virus-induced inflammation model for investigating the anti-inflammatory effects of *A. camphorata*. As shown in Appendix A, the methanolic extract of *A. camphorata* mycelium (ACM) showed significant inhibition activity against excessive NO production in poly I:C-induced RAW 264.7 cells. The percentages of inhibition of NO production treated with ACM at concentrations of 5, 10, 20, and 40 μg/mL were 53.39 ± 4.87, 67.83 ± 5.74, 84.35 ± 1.51, and 90.15 ± 1.33%, respectively. The methanolic extract of *A. camphorata* mycelium was then subjected to MPLC separation, and each of the collected fractions was assigned to one of ten main fractions based on TLC characterization. Fraction 5 potently inhibited NO production with percentages of inhibition of 42.00 ± 2.86, 51.46 ± 5.62, 69.05 ± 6.25, and 78.58 ± 3.68% at concentrations of 1, 2, 5, and 10 μg/mL, respectively. Thus, Fraction 5 was chosen for further analysis.

The ^1^H NMR experiment offers metabolite profiling, achieved by chemical shifts, *J*-couplings, and signal area integration. Particularly, the intensity of the signal in ^1^H NMR spectra is proportional to the relative number of (equivalent) protons that give rise to the signal and thus could provide quantitative information about the different compounds in the mixture. Relating putative bioactivities with these metabolites is feasible by integrating metabolite profiling with bioactivity results. According to the inspection of ^1^H NMR spectra, the presence of 4-acetylantroquinonol B (**1**) in Fraction 5 was deduced from the characteristic signals as follows: A highly deshielded proton signal at *δ*_H_ 5.72 (1H, d, *J* = 3.2 Hz) assigned as H-4; two olefinic proton signals at *δ*_H_ 5.21 and 5.11 (1H each, *t*, *J* = 7.0 Hz); an oxygenated proton signal at *δ*_H_ 4.62 (1H, m) assigned as H-15; two methoxy proton signals at *δ*_H_ 4.00 and 3.67 (3H each, s); acetyl proton signal at *δ*_H_ 2.09 (3H, s); two methyl proton singlets at *δ*_H_ 1.65 and 1.55 (3H each, s); and one methyl proton doublet at *δ*_H_ 1.19 (3H, d, *J* = 7.3 Hz). The remaining methyl proton doublet at *δ*_H_ 1.26 was overlapped. The relative abundance of different compounds in Fraction 5 obtained by comparing the signal intensity in the ^1^H NMR spectrum (Figure 1) then indicated that the anti-inflammation activity might be contributed to by the presence of 4-acetylantroquinonol B (**1**), which is a natural ubiquinone analogue isolated from the mycelium of *A. camphorata* in 2009 [22]. As one of the bioactive compounds in *A. camphorata*, it has been reported to possess various bioactivities, which include anti-inflammatory, anti-carcinogenic, hepatoprotective, and immunomodulatory properties [22,23,24,25,26,27,28]. However, the role of 4-acetylantroquinonol B (**1**) in a viral-induced inflammation model has not been reported previously.

To assess the anti-inflammation activity of compound **1** and other phytochemicals in the bioactive Fraction 5, it was further isolated and purified by HPLC. In addition to the identification of 4-acetylantroquinonol B (Appendix A), the phytochemical investigation of Fraction 5 yielded seven compounds, including two antroquinonols, antroquinonol Y (**2**) [29] and antroquinonol (**3**) [30]; a benzenoid, 3-hydroxy-4,5-dimethoxy-2-methylbenzaldehyde (**4**); two maleimide derivatives, 3-isobutyl-4-(4-methoxyphenyl)-1*H*-pyrrol-1-ole-2,5-dione (**5**) and 3-(4-hydroxyphenyl)-4-isobutyl-1H-pyrrole-2,5-dione (**6**) [31]; and two maleic anhydride derivatives, antrocinnamomin C (**7**) [32] and antrocinnamomin D (**8**) [32]. The structures of compounds **1**–**8** are shown in Figure 2. Their structures were identified by spectral analysis and comparing their data to those reported in the literature. Among them, compounds **4** and **5** were identified as new compounds. Detailed structural elucidation of compounds **4** and **5** was further illustrated by spectral analysis, including MS, IR, and 1D and 2D NMR (Appendix A).

Compound **4** was determined to have the molecular formula of C_10_H_12_O_4_ with five degrees of unsaturation according to its HRESIMS peak at *m*/*z* 197.0812 [M + H]^+^. The IR spectrum showed peaks at 3437 (OH), 1670 (aldehyde), and 1593 and 1493 (aromatic) cm^−1^. The ^1^H and ^13^C NMR data revealed the presence of a benzene ring with five substituents—an aldehyde (*δ*_H_ 10.25, s), an aryl proton (*δ*_H_ 7.02, s, 1H), a hydroxyl proton (*δ*_H_ 6.03, s, 1H), two methoxy groups (*δ*_H_ 3.99 and 3.90, s, 3H each), and a deshielded methyl group (*δ*_H_ 2.51, s, 3H), together with six aromatic carbon signals (*δ*_C_ 150.1, 148.0, 140.1, 129.7, 121.4, and 105.0). The methyl signal (*δ*_H_ 2.51) resonated at a higher frequency, then indicated that the methyl group is located at the *ortho*-position to the aldehyde group. The relative positions of aryl substituents were confirmed by HMBC correlations (Figure 3) from the methyl protons (*δ*_H_ 2.51) to C-1 (*δ*_C_ 129.7), C-2 (*δ*_C_ 121.4), and C-3 (*δ*_C_ 148.0), from the aldehyde proton (H-1′, *δ*_H_ 10.25) to C-1 (*δ*_C_ 129.7) and C-6 (*δ*_C_ 105.0), and from the aryl proton (H-6, *δ*_H_ 7.02) to C-2 (*δ*_C_ 121.4), C-4 (*δ*_C_ 140.1), and C-1′ (*δ*_C_ 191.2). Thus, compound **4** was identified as 3-hydroxy-4,5-dimethoxy-2-methylbenzaldehyde.

Compound **5** was determined to have the molecular formula of C_15_H_17_NO_4_ with eight degrees of unsaturation according to its HRESIMS peak at *m*/*z* 298.1051 [M + Na]^+^. The IR spectrum showed peaks at 3312 (N-OH), 1776 and 1709 (imide), and 1607 and 1514 (aromatic) cm^−1^. The ^1^H NMR data showed the presence of a *para*-substituted aromatic ring (*δ*_H_ 7.47 (d, *J* = 8.6 Hz, 2H) and 6.92 (d, *J* = 8.6 Hz, 2H)), a hydroxy group (*δ*_H_ 5.55, s, 1H), a methoxy group (*δ*_H_ 4.00, s, 3H), and an isobutyl group (*δ*_H_ 2.51 (d, *J* = 7.4 Hz, 2H), 2.05 (m, 1H), and 0.90 (d, *J* = 6.7 Hz, 6H)). The COSY correlations between H-2′ and H-1′ and two methyl groups further confirmed the presence of the isobutyl group. The ^13^C NMR data revealed the presence of two carbonyl carbons (*δ*_C_ 167.8 and 167.1) and two quaternary olefinic carbons (*δ*_C_ 136.3 and 135.9), which was characteristic of a maleimide moiety. The relative connectivity was further deduced from the HMBC correlations from H-1′ (*δ*_H_ 2.51) to C-2 (*δ*_C_ 167.8) and C-3 (*δ*_C_ 136.3), and the HMBC correlations (Figure 3) from H-2′′ (*δ*_H_ 7.47) to C-4 (*δ*_C_ 135.9) and C-4′′ (*δ*_C_ 157.4). The overall spectroscopic data of compound **5** were similar to those of antrocinnamomin B [32]. The only difference is the presence of a methoxy group (*δ*_C_/*δ*_H_ 65.9/4.00) in compound **5** instead of a hydroxy group in antrocinnamomin B. Thus, compound **5** was identified as 3-isobutyl-4-(4-methoxyphenyl)-1*H*-pyrrol-1-ol-2,5-dione.

Evaluations for the inhibitory effect on NO production in RAW 264.7 macrophages induced by poly I:C (Table 1) were performed on all isolated compounds. In order to exclude the cytotoxic effect of tested compounds, those were also applied for cell viability assessment. No cytotoxicity was observed (cell viability >90%) when treated with the tested compounds under the half maximal inhibitory concentration. As expected, 4-acetylantroquinonol B (**1**) shows most potent inhibitory activity against poly I:C induced-NO production in RAW 264.7 cells with an IC_50_ value of 0.57 ± 0.06 μM. Additionally, two antroquinonols (**2** and **3**), a benzenoid (**4**), and two maleimide anhydride derivatives (**5** and **6**) showed moderate inhibitory activity with the IC_50_ values ranging from 2.96 to 27.14 μM. However, maleic anhydride derivatives (**7** and **8**), compared to maleimide derivatives, were found to be not effective in this study. According to the bioactivity results above, together with the relative abundance of different metabolites in the bioactive Fraction 5, 4-acetylantroquinonol B (**1**) was not only the most abundant compound but the most potent compound against poly I:C-induced NO production in RAW 264.7 cells.

Toll-like receptors (TLRs), a class of PPR, play a crucial role in regulating the innate immune system. TLR3 could recognize dsRNA and drive the inflammatory cytokine signaling, such as nuclear factor-κB (NF-κB) and mitogen-activated protein (MAP) kinase activation [19]. The preliminary activity of ACM, together with the anti-inflammatory compounds in this study, urges the need for further studies on the TLR3 signaling pathway.

## 3. Materials and Methods

### 3.1. General Experimental Procedures

Silica gel (25 μm) was used for medium-pressure column chromatography (MPLC). Silica gel 60 F254 plates (200 μm) was used for thin-layer chromatography (TLC). High-performance liquid chromatography (HPLC) was carried out on a KNAUER HPLC system equipped with a refractive index (RI) detector. IR spectra were measured with a Shimadzu IRPrestige-21 Fourier transform infrared (FT-IR) spectrophotometer. High-resolution electrospray mass (HRESIMS) spectra were acquired on a Bruker Daltonics maXis impact ESI-Q-TOF mass spectrophotometer. UV spectra were obtained using a PerkinElmer Lambda 265 UV–Visible spectrophotometer. Nuclear magnetic resonance (NMR) spectra were measured with a Bruker AVANCE NEO 400 MHz FTNMR spectrometer and a Bruker AVANCE 500 MHz FTNMR spectrophotometer.

### 3.2. Chemicals

Acetone, dichloromethane, ethyl acetate (EtOAc), *n*-hexane, and deuterated chloroform were obtained from the branch of Merck in Taiwan. Antibiotics (penicillin-streptomycin), Dulbecco’s modified Eagle’s medium (DMEM), glutamine, and fetal bovine serum (FBS) were purchased from GIBCO (Grand Island, NY, USA). Indomethacin, N-(1-naphthyl)ethylenediamine dihydrochloride, polyinosinic-polycytidylic acid (Poly I:C), potassium, phosphate-buffered saline (PBS), sodium nitrite, sulphanilamide, and phosphoric acid were obtained from Sigma Chemical Co. (St. Louis, MO, USA).

### 3.3. Extraction and Isolation

The mycelium of *A. camphorata* (4.1 kg) was extracted with methanol three times at room temperature, and the crude extract was concentrated and applied on MPLC with a linear gradient solvent system of *n*-hexane and EtOAc. Each collected fraction was assigned into ten main fractions based on TLC characterization.

All compounds were obtained from Fraction 5 by semi-preparative HPLC, equipped with a refractive index (RI) detector at a flow rate of 3 mL/min. Fraction 5 was separated by HPLC to yield compound **1** (801.1 mg, *n*-hexane/EtOAc = 2/1, *t*_R_ = 18.6 min), re-separated by HPLC (dichloromethane/EtOAc = 6/1) to provide compounds **6** (3.2 mg, *t*_R_ = 14.1 min), **7** (2.6 mg, *t*_R_ = 6.6 min), and **8** (1.6 mg, *t*_R_ = 8.3 min), and further purified by HPLC to give compounds **5** (5.1 mg, *n*-hexane/acetone = 4/1, *t*_R_ = 15.2 min), **2** (43.6 mg, *n*-hexane/EtOAc = 4/1, *t*_R_ = 23.3 min), **3** (mg, *n*-hexane/EtOAc = 3/2, *t*_R_ = 12.5 min), and **4** (7.2 mg, *n*-hexane/EtOAc = 4/1, *t*_R_ = 18.1 min).

3-Hydroxy-4,5-dimethoxy-2-methylbenzaldehyde (4): Amorphous colorless powder; IR (KBr) *ν*_max_ 3437, 3001, 2940, 2866, 1670, 1593, and 1493 cm^−1^; UV *λ*_max_ (log *ε*, MeOH) 286 (3.81); ^1^H and ^13^C NMR data: Shown in Table 2; HRESIMS *m*/*z* 197.0812 [M + H]^+^.

3-Isobutyl-4-(4-methoxyphenyl)-1H-pyrrol-1-ol-2,5-dione (5): Amorphous colorless powder; IR (KBr) *ν*_max_ 3312, 2955, 2924, 2866, 1776, 1709, 1607, 1580, and 1514 cm^−1^; UV *λ*_max_ (log *ε*, MeOH) 232 (3.63), 284 (3.00), 370 (2.87); ^1^H and ^13^C NMR data: Shown in Table 3; HRESIMS *m*/*z* 298.1051 [M + Na]^+^.

### 3.4. NO Production and Cell Viability of RAW 264.7 Macrophages Induced by Poly I:C

RAW 264.7 macrophages were cultured in DMEM supplemented with 10% FBS and 1% antibiotics in an incubator with a humidified atmosphere of 5% CO_2_ at 37 °C. After being seeded into 96-well plates (5 × 10^4^ cells/well), the cells were incubated overnight and then treated with filtered, tested samples with different concentrations, followed by poly I:C (50 μg/mL). After incubation for 24 h, NO concentration in the culture medium was measured with the Griess reaction and calculated using a standard curve. Each supernatant (100 μL) was reacted with the same volume (100 μL) of Griess reagent (containing 0.05% *N*-(1-naphthyl)ethylenediamine dihydrochloride, 0.5% sulphanilamide, and 2.5% phosphoric acid) for 5 min at room temperature. The absorbance at 540 nm was detected using a microplate reader. The CCK-8 assay was applied for cell viability assessment according to standard protocols. The absorbance at 450 nm was detected using a microplate reader. The experiments were performed in triplicates.

### 3.5. Statistical Analysis

The results were expressed as mean ± SD of triplicates. Significant differences were examined using a two-tailed unpaired Student’s *t* test. *p* values of less than 0.05 were considered statistically significant.

## 4. Conclusions

In this study, the inhibitory effects of the methanolic extract of the mycelium of *A. camphorata* were assessed on viral infection-associated inflammation using poly I:C-stimulated NO production in a RAW 264.7 mouse macrophage model. Our study provides a precise strategy for screening anti-inflammatory compounds from the mycelium of *A. camphorata* by a combination of bioactivity-guided isolation with an NMR-based identification approach, which led to isolation and identification of eight compounds, including two new compounds **4** and **5**. Among them, 4-acetylantroquinonol B (**1**) was not only the most abundant compound, but the most potent compound in the bioactive Fraction 5 from the mycelium of *A. camphorata*. Thus, it could be a chemical marker for the potential anti-inflammatory agent *A. camphorata* mycelium during viral infection.

## Figures and Tables

**Figure 1 molecules-27-05320-f001:**
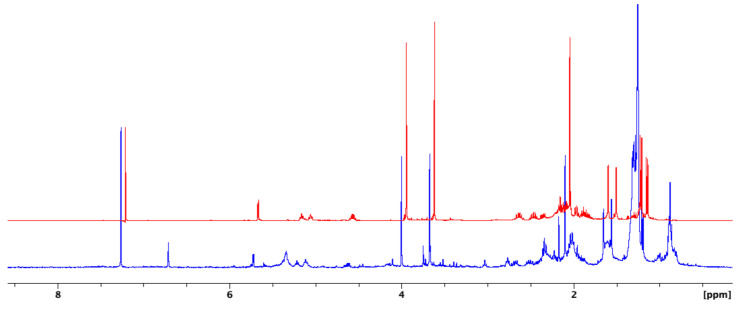
Stacked ^1^H NMR spectra of the bioactive Fraction 5 (blue) and the major compound 4-acetylantroquinonol B (red).

**Figure 2 molecules-27-05320-f002:**
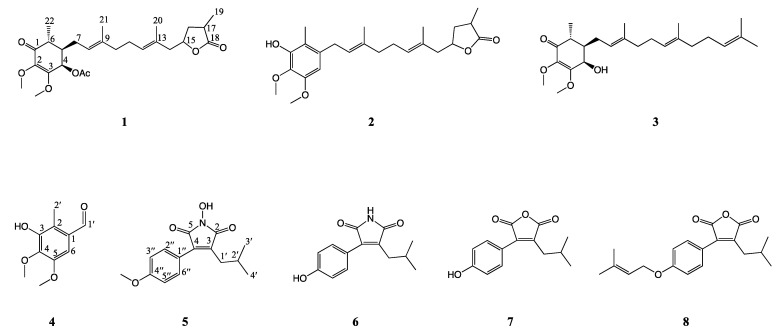
The structures of compounds **1**–**8** from the mycelium of *A. camphorata*.

**Figure 3 molecules-27-05320-f003:**
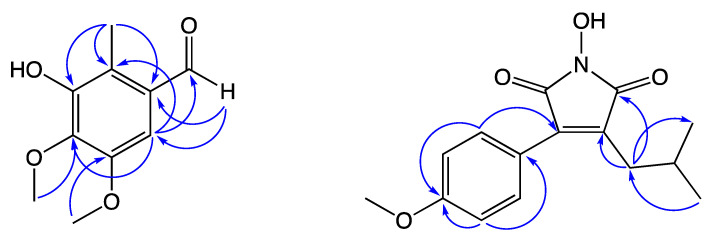
Key HMBC correlations of compounds **4** and **5**.

**Table 1 molecules-27-05320-t001:** Anti-inflammatory activity of compounds **1**–**8** from *A. camphorata* on NO production in RAW 264.7 macrophages induced by poly I:C.

Compounds	IC_50_ (μM) ^1^
**1**	0.57 ± 0.06
**2**	2.96 ± 0.77
**3**	10.80 ± 1.32
**4**	10.11 ± 1.76
**5**	11.30 ± 3.34
**6**	27.14 ± 4.61
**7**	>50
**8**	>50
Indomethacin ^2^	67.10 ± 2.23

^1^ The IC_50_ values were calculated based on the dose–response curves and expressed as mean ± SD of triplicates. ^2^ Indomethacin was used as a positive control.

**Table 2 molecules-27-05320-t002:** ^13^C (100 MHz) and ^1^H NMR (400 MHz, Chloroform-*d*), and HMBC data of compound **4.**.

Position	^13^C NMR *δ*_C_	^1^H NMR *δ*_H_ (Multiplicity)	HMBC
1	129.7		
2	121.4		
3	148.0		
4	140.1		
5	150.1		
6	105.0	7.02 (s)	C-2, C-4, C-5, C-1′
1′	191.2	10.25 (s)	C-1, C-6
2′	9.93	2.51 (s)	C-1, C-2, C-3
3-OH		6.03 (s)	C-2, C-3

**Table 3 molecules-27-05320-t003:** ^13^C (125 MHz) and ^1^H NMR (500 MHz, Chloroform-*d*), and HMBC data of compound **5.**.

Position	^13^C NMR *δ*_C_	^1^H NMR *δ*_H_ (Multiplicity)	HMBC
2	167.8		
3	136.3		
4	135.9		
5	167.1		
1′	33.1	2.51 (d, 7.4)	C-2, C-3, C-3′, C-4′
2′	28.2	2.05 (m)	C-1′, C-3′, C-4′
3′, 4′	22.9	0.90 (d, 6.7)	C-1′, C-2′
1″	121.2		
2″, 6″	131.5	7.47 (d, 8.6)	C-4, C-3″, C-4″, C-5″
3″, 5″	115.9	6.92 (d, 8.6)	C-1″, C-4″
4″	157.4		
4″-OCH_3_	65.9	4.00 (s)	
OH		5.55 (brs)	

## Data Availability

The data presented in this study are available in the Appendix A.

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
