# Peer review of "Anti-Inflammatory Constituents of *Antrodia camphorata* on RAW 264.7 Cells Induced by Polyinosinic-Polycytidylic Acid"

_molecules, 2022, doi:10.3390/molecules27165320_

Round 1

Reviewer 1 Report

Authors isolated 8 compounds from a methanolic extract of the taiwanese endemic mycelium Antrodia Camphorata. Among them the most abundand was 4-acetylantroquinonol B named as compound 1 which is a known bioactive agent since it shows anti-inflammatory, anti-carcinogenic, hepatoprotective, and immunomodulatory properties. All 8 isolated coumpounds were tested for their inhibitory effect on "in vitro" viral infection induced NO production in RAW 264.7. As expected the most inhibiting effect was shown by 4-acetylantroquinonol B. The study is interesting and experiments related to isolation and characterization of the 8 compound  are  convincing, however major issues remain to be solved.

Introduction

- Why do the authors specifically refer to COVID19 in the introduction? Covid 19 is one of the viral infections that could be benefit by anti inflammatory drugs treatment that reduce oxidative stress. The introduction should be reworked by taking a broader vision at this issue.

Methods

-It's not clear how biological evaluations were carried out for both cytotoxicity test and the assay of NO production. More specific information needs to be given.

-What statistical analysis was performed? In Fig. 1S there are "statistical signs" but no comment is reported in legend.

Results

-What about the cytotoxicity test? The results obtained are not reported and/or commented in the manuscript.

-I would like to see the raw data of the inhibitory effect of the 8 compound on NO production. 

-Why did the authors choose indomethacin as positive control instead of a specific inhibitor of nitric oxide synthase enzyme?

Author Response

The authors appreciate your suggestions that help improve our manuscript. Following are the reviewer comments with our response in blue.

Introduction

- Why do the authors specifically refer to COVID19 in the introduction? Covid 19 is one of the viral infections that could be benefit by anti inflammatory drugs treatment that reduce oxidative stress. The introduction should be reworked by taking a broader vision at this issue.
Introduction was revised based on this suggestion. Please see Line 25-40.

Methods

-It's not clear how biological evaluations were carried out for both cytotoxicity test and the assay of NO production. More specific information needs to be given.
Method (NO production and Cell Viability) were revised. Please see Line 415-427.

-What statistical analysis was performed? In Fig. 1S there are "statistical signs" but no comment is reported in legend.
Statistical analysis was added. Please see Line 428-431.

Results

-What about the cytotoxicity test? The results obtained are not reported and/or commented in the manuscript.
The concentration unit of μg/mL was performed in the bioassessments but was converted to μM after calculation of IC50. Thus, the cytotoxicity data was not shown.

The description ‘‘No cytotoxicity was observed ...’’ was added in Line 752.

-I would like to see the raw data of the inhibitory effect of the 8 compound on NO production. 
We are willing to provide our raw data, but that could not be upload in submission system. We'll provide raw data to editor.

-Why did the authors choose indomethacin as positive control instead of a specific inhibitor of nitric oxide synthase enzyme?
The positive control was selected based on the references 3 and 21.

Reviewer 2 Report

The Manuscript molecules-1828108-peer-review-v1

Manuscript molecules-1828108-peer-review-v1 describes the anti-inflammatory constituents of Antrodia camphorate on Polyinosinic-Polycytidylic Acid-Induced NO production in RAW 264.7 Cells.

The manuscript has important findings that could be of interest to the readers of Molecules. However, the following comments need to be addressed before publication.    

Major concerns:

1.       The introduction part needs to be rewritten in a way to elucidate the gap in knowledge that the current study is trying to explore.  

2.       Introduction, lines 25-45: the first two paragraphs of the introduction section are unrelated to the topic of the manuscript. SARS-CoV-2 is known to be a single-stranded RNA virus. Therefore, the use of poly I:C (double-stranded RNA analog) as an inducer of inflammation cannot be correlated to the inflammatory processes induced by SARS-CoV-2.

3.       Materials and Methods: The authors have used a statistical analysis to show significance as indicated in Figure S1. However, the type and method of analysis were not described in the methods section. Also, it is unclear what expression was used to describe the results, was it mean ± SD? This needs to be clarified in the methods section.

4.       Results and Discussion, Lines 138-139: “Among the tested fractions, Fraction 5 significantly inhibited NO production” the authors need to show the results of NO inhibition for other fractions to assume that Fraction 5 was the most active one. Also, it is unclear which concentration of fraction 5 has caused significant inhibition and at what significant level?

5.       Results and Discussion, Lines 154-157: the authors have assumed that the abundance of 4-acetylantroquinonol B (1), based on signal intensity in the 1H NMR spectrum, indicated that the anti-inflammatory activity is correlated to compound (1). Such a proposition can’t be concluded based on the relative abundance of a compound in a mixture unless it is combined with the anti-inflammatory effect of each component. Although authors have already examined the anti-inflammatory effect of each component alone, it seems that they jumped to a premature conclusion in this particular part of the manuscript. It is recommended to postpone such a proposition to the end of the manuscript.

Minor concerns:

1.       Introduction, line 50: the following sentence “ ……… anti-aging, anti-carcinogenic, immunomodulatory, hepatoprotective, cardioprotective, and neuroprotective effects.” needs reference(s).

2.       Abbreviations: the abbreviated words and phrases must be mentioned in full as they first appear in the text. For instant, dsRNA has first appeared in the introduction while the full term was described in the results and discussion. Similar note was observed for MPLC. Other abbreviations were not described in the manuscript, such as EtOAc.  

Author Response

The authors appreciate your suggestions that help improve our manuscript. Following are the reviewer comments with our response in blue.

Major concerns:

  1. The introduction part needs to be rewritten in a way to elucidate the gap in knowledge that the current study is trying to explore.  
  2. Introduction, lines 25-45: the first two paragraphs of the introduction section are unrelated to the topic of the manuscript. SARS-CoV-2 is known to be a single-stranded RNA virus. Therefore, the use of poly I:C (double-stranded RNA analog) as an inducer of inflammation cannot be correlated to the inflammatory processes induced by SARS-CoV-2.
    Introduction was revised with a broader vision at this issue. 
  3. Materials and Methods: The authors have used a statistical analysis to show significance as indicated in Figure S1. However, the type and method of analysis were not described in the methods section. Also, it is unclear what expression was used to describe the results, was it mean ± SD? This needs to be clarified in the methods section.
    Statistical analysis was added according to this suggestion. 
  4. Results and Discussion, Lines 138-139: “Among the tested fractions, Fraction 5 significantly inhibited NO production” the authors need to show the results of NO inhibition for other fractions to assume that Fraction 5 was the most active one. Also, it is unclear which concentration of fraction 5 has caused significant inhibition and at what significant level?
    The description was revised.
  5. Results and Discussion, Lines 154-157: the authors have assumed that the abundance of 4-acetylantroquinonol B (1), based on signal intensity in the 1H NMR spectrum, indicated that the anti-inflammatory activity is correlated to compound (1). Such a proposition can’t be concluded based on the relative abundance of a compound in a mixture unless it is combined with the anti-inflammatory effect of each component. Although authors have already examined the anti-inflammatory effect of each component alone, it seems that they jumped to a premature conclusion in this particular part of the manuscript. It is recommended to postpone such a proposition to the end of the manuscript.
    As shown in Figure 1, compound 1 was quite major in Fraction 5. The contribution of compound 1 was ‘‘indicated’’ and confirmed by testing its activity on NO production. Please see Line 730-731.
    The description was revised.

Minor concerns:

  1. Introduction, line 50: the following sentence “ ……… anti-aging, anti-carcinogenic, immunomodulatory, hepatoprotective, cardioprotective, and neuroprotective effects.” needs reference(s).
    The references were added.
  2. Abbreviations: the abbreviated words and phrases must be mentioned in full as they first appear in the text. For instant, dsRNA has first appeared in the introduction while the full term was described in the results and discussion. Similar note was observed for MPLC. Other abbreviations were not described in the manuscript, such as EtOAc.
     The abbreviations were revised.

Round 2

Reviewer 1 Report

The authors have improved their manuscript as suggested. In my opinion, in this form, the manuscript could be accepted.

Reviewer 2 Report

The revised version of the manuscript was improved.